# Butyrate Protects against SARS-CoV-2-Induced Tissue Damage in Golden Hamsters

**DOI:** 10.3390/ijms241814191

**Published:** 2023-09-16

**Authors:** Huan Yu, Lunzhi Yuan, Zhigang Yan, Ming Zhou, Jianghui Ye, Kun Wu, Wenjia Chen, Rirong Chen, Ningshao Xia, Yi Guan, Huachen Zhu

**Affiliations:** 1Guangdong-Hong Kong Joint Laboratory of Emerging Infectious Diseases/Joint Laboratory for International Collaboration in Virology and Emerging Infectious Diseases (Key Laboratory of Ministry of Education), Joint Institute of Virology (Shantou University/The University of Hong Kong), Shantou University Medical College, Shantou 515063, China; 2State Key Laboratory of Vaccines for Infectious Diseases, National Institute of Diagnostics and Vaccine Development in Infectious Diseases, NMPA Key Laboratory for Research and Evaluation of Infectious Disease Diagnostic Technology, School of Public Health, Xiamen University, Xiamen 361102, China; 3State Key Laboratory of Emerging Infectious Diseases (SKLEID), School of Public Health, Li Ka Shing Faculty of Medicine, The University of Hong Kong, Hong Kong SAR, China; 4EKIH (Gewuzhikang) Advanced Pathogen Research Institute, Futian District, Shenzhen 518045, China

**Keywords:** butyrate, SARS-CoV-2, golden hamster, type I IFN, apoptosis, oxidative stress

## Abstract

Butyrate, produced by gut microbe during dietary fiber fermentation, has anti-inflammatory and antioxidant effects on chronic inflammation diseases, yet it remains to be explored whether butyrate has protective effects against viral infections. Here, we demonstrated that butyrate alleviated tissue injury in severe acute respiratory syndrome coronavirus 2 (SARS-CoV-2)-infected golden hamsters supplemented with butyrate before and during the infection. Butyrate-treated hamsters showed augmentation of type I interferon (IFN) response and activation of endothelial cells without exaggerated inflammation. In addition, butyrate regulated redox homeostasis by enhancing the activity of superoxide dismutase (SOD) to inhibit excessive apoptotic cell death. Therefore, butyrate exhibited effective prevention against SARS-CoV-2 by upregulating antiviral immune responses and promoting cell survival.

## 1. Introduction

Short-chain fatty acids (SCFAs) are the most abundant metabolites, mainly produced by the gut microbiota in the colon via fermentation of dietary fiber [1,2]. Among SCFAs, butyrate is a primary energy source for colonocytes and a well-known anti-inflammatory mediator which can be activated by binding to G protein-coupled receptors (GPRs), mainly GPR41 and GPR43 [3], or can inhibit the activity of histone deacetylase (HDAC) [4]. Butyrate can not only regulate mucosal barrier function and mucosal immunity, but also mediate the communication between colonic microbiota and other organs such as brain, lung and liver [5,6,7,8].

Lower respiratory infections are reportedly the fourth leading cause of death worldwide, with 2.6 million global deaths in 2019 [9]. Since December 2019, severe acute respiratory syndrome coronavirus 2 (SARS-CoV-2) has caused more than 6.9 million deaths as of August 2023 [10]. SARS-CoV-2 can replicate both in the upper and lower respiratory tract [11,12]. SARS-CoV-2 infection, alongside individual susceptibility and host immunity, can even progress to severe and life-threatening pneumonia, which is responsible for increased morbidity and mortality in Coronavirus Disease 2019 (COVID-19) [13]. Patients with either acute COVID-19 or post-acute COVID Syndrome (PACS) were reported to have gastrointestinal symptoms such as abdominal pain, diarrhea, nausea and vomiting [14,15]. Interestingly, patients with PACS at 6 months showed gut microbiome dysbiosis compared with non-COVID-19 controls and patients without PACS, while the development of PACS was not significantly correlated with viral load both in respiratory and stool [16]. Therefore, there is growing emphasis on how butyrate may maintain intestinal homeostasis and reduce lung disruption in SARS-CoV-2 infection. 

So far, many therapeutic approaches have been developed for COVID-19, including the use of antiviral drugs, monoclonal antibodies, immunomodulators, and convalescent plasma [17,18,19,20,21]. However, there are limited studies about whether the interventions based on gut microbes and metabolites are effective for the prevention and treatment of respiratory viral infection. In vitro studies have observed that butyrate can regulate the expression of genes involved in the antiviral pathway [22,23,24]. We hypothesized that oral administration of butyrate protects against SARS-CoV-2 infection. Here, we investigated the effects of butyrate on the colon mucosal barrier and lung injury in SARS-CoV-2-challenged golden hamsters. The results showed that butyrate can significantly increase the number of goblet cells in the colon. More importantly, supplementation with butyrate boosted antiviral immune responses and promoted cell survival in the lung, and as a result, alleviated lung injury of SARS-CoV-2-infected hamsters. 

## 2. Results

### 2.1. Body Weight Change and Viral Load in Hamsters

To assess whether microbial metabolites protect against virus-induced inflammation and tissue injury, golden hamsters were orally administrated butyrate before and during the course of SARS-CoV-2 infection (Figure 1A). From 2 to 5 days post-inoculation (dpi), hamsters in the virus-inoculated groups, either butyrate-treated or untreated, showed a significant decrease in body weight, while the mock-infected group showed slight body weight gain. There was no significant difference between the control and the butyrate-treated groups in weight loss (Figure 1B). To further investigate the effect of butyrate on SARS-CoV-2 replication, we assessed the viral RNA and viral titer in the trachea, lungs and colon of the hamsters. The highest viral RNA and viral titer was detected in the lungs of virus-inoculated individuals throughout the infection, while the colon had the lowest viral RNA and viral titer (Figure 1C,D). The mock-infected group had no virus infection. From 3 to 5 dpi, viral titer in the trachea and lung was significantly decreased both in the control and butyrate-treated hamsters (Figure 1D). No statistical significance was observed in the viral RNA and viral titer between butyrate-treated hamsters and the control (*p* > 0.05; Figure 1C,D).

### 2.2. Pulmonary Pathological Changes in Hamsters

At 5 dpi, gross observation showed pulmonary hemorrhage and edema in 50–75% of control hamsters, and in 10–50% of butyrate-treated hamsters (Figure 2A). Histopathological analysis of the lung revealed that hamsters treated with butyrate had significantly lower pathological scores (*p* < 0.001) with less inflammatory cell infiltration, reduced alveolar structure damage, and less hemorrhage at 5 dpi (Figure 2B,C). Immunohistochemistry (IHC) for SARS-CoV-2 nucleocapsid protein (NP) detection indicated that specific viral antigens were widely expressed and distributed in the alveolar cells and pulmonary exudate of each control animal. In contrast, NP-positive cells were majorly observed in the epithelial cells of bronchioles in the butyrate-treated hamsters (Figure 2B). These distinct viral antigen distribution patterns (Figure 2B), rather than differences in the viral load (Figure 1C,D), suggested limited virus dissemination in the bronchioloalveolar tissues following butyrate treatment, which in turn protected against SARS-CoV-2 by reducing tissue destruction (Figure 2A,B) and overall pathology (Figure 1B).

### 2.3. Expression Levels of Representative Genes in Hamsters

To elucidate how butyrate influences antiviral innate immunity, we assessed the expression levels of genes involved following SARS-CoV-2 infection (see Appendix A for primer sequences). First, a marked reduction in interferon alpha and beta receptor subunit 1 gene (*Ifnar1*) was observed in the control compared with both mock and butyrate-treated hamsters at 5 dpi, indicating an inhibited type I interferon (IFN) signaling induced by SARS-CoV-2 (Figure 3A). Inflammatory cytokine genes such as interleukin 6 (*Il6*), *Il1b* and tumor necrosis factor alpha (*Tnfa*) were both upregulated in two virus-inoculated groups, with higher levels in butyrate-treated hamsters (Figure 3B). The same was true of the proinflammatory interferon gamma gene (*Ifng*) (Figure 3D). Next, to test whether butyrate can influence the immunomodulatory functions of endothelial cells, we assessed mRNA expression of cellular adhesion molecules. Intercellular adhesion molecule 1 gene (*Icam1*) was marginally downregulated in the control compared with the mock, whereas increased expression was observed in butyrate-treated hamsters (Figure 3C). Vascular cell adhesion molecule 1 (*Vcam1*) and selectin E (*Sele*) genes were both upregulated after SARS-CoV-2 stimulation, but no significant upregulation was seen in *Vcam1* between mock and control hamsters (Figure 3C). Determination of these adhesion molecules showed that endothelial cells in the lung of butyrate-treated hamsters were activated at 5 dpi. Finally, we found endothelial nitric oxide synthase (eNOS, *Nos3*) and inducible NOS (iNOS, *Nos2*) were significantly decreased in the control compared with the mock, indicating deficient nitric oxide (NO) inside blood vessels upon infection, which thus leads to endothelial dysfunction and suppressed NO signaling in regulating inflammation (Figure 3D). However, butyrate reversed the downregulation of these two nitric oxide synthases in the lung at 5 dpi (Figure 3D). Taken together, these results showed that butyrate regulated inflammation by activating antiviral response and promoting homeostasis and the activation of endothelial cells.

### 2.4. Oxidative Status in Hamsters

To further determine the pathogenesis of inflammation, we assessed oxidative stress at 5 dpi. The expression level of NADPH oxidase 2 gene (*Nox2*) showed a slight increase in the control and butyrate-treated hamsters, which probably pointed toward the production of reactive oxygen species (ROS) (Figure 4A). Compared with the mock, the level of malondialdehyde (MDA) in the plasma of the control was markedly elevated, indicating lipid peroxidation subsequent to oxidative stress (Figure 4B). Moreover, the activity of superoxide dismutase (SOD), a key antioxidant enzyme in redox signaling, was significantly decreased in the control compared with butyrate-treated hamsters (Figure 4C). Therefore, butyrate contributed to anti-inflammatory effects through reduction of oxidative stress, mainly by regulating redox signaling.

### 2.5. Apoptosis in Hamsters

To determine the consequences of oxidative stress and whether structural integrity of the lung was also affected by butyrate, we assessed SARS-CoV-2-induced apoptosis at 5 dpi. Hoechst staining showed that the lung of butyrate-treated hamsters compared with the control had significantly fewer apoptotic cells (Figure 5A,B). In line with this, caspase-8 (*Casp8*), a crucial initiator in apoptotic pathway, was significantly increased in the control, but there was no significant difference between mock and butyrate-treated hamsters (Figure 5C). Executioner caspase, especially *Casp3*, was upregulated in virus-inoculated groups either butyrate-treated or untreated, indicating apoptosis upon SARS-CoV-2 infection (Figure 5C). However, the expression of *Bcl2*, an antiapoptotic signature gene, showed a significant increase when hamsters were treated with butyrate (Figure 5C). Thus, butyrate alleviated lung injury by preventing excessive apoptotic cell death and promoting cell survival mediated by an antiapoptotic gene.

### 2.6. Goblet Cells and Muc2 Expression in Hamsters

As butyrate is the metabolite mainly produced in colon, we also assessed whether butyrate regulated the development of the mucosal barrier. Compared with butyrate-treated hamsters, there were significantly fewer goblet cells in the colon of the control (Figure 6A,B). Similarly, crypts were elongated (*p* < 0.001), and mucin 2 (*Muc2*) expression was somewhat increased (*p* > 0.05) in butyrate-treated hamsters (Figure 6B,C). Thus, SARS-CoV-2 infection impaired the colon mucosal barrier and butyrate played a role in goblet cell development. 

## 3. Discussion

Here, we demonstrated that butyrate could protect SARS-CoV-2-infected hamsters by enhancing antiviral response and promoting cell survival to maintain tissue homeostasis. In severe and critical COVID-19 patients, low or no type I IFNs levels were observed, suggesting a highly impaired type I IFN response in these patients [25,26]. Similarly, the expression of IFNB1 and IFN-stimulated genes (ISGs) such as *Mx1*, *Isg20* and *Oasl* failed to be activated in SARS-CoV-2-infected golden hamsters and ferrets, respectively [26,27]. At the same time, IFN-I/II receptor-double-knockout mice had increased viral titers and higher congestion scores of the lungs following SARS-CoV-2 infection [28]. We found that butyrate activated an innate immune response in the early phase of SARS-CoV-2 infection, characterized by upregulated type I IFN signaling and increased proinflammatory cytokines, which contributed to rapid viral antigen clearance and avoided immunopathology in the lungs. This was also supported by a recent study which showed that antiviral factors such as IL1b, IRF7, TNF and IFNAR1 were upregulated in butyrate-treated gut epithelial organoids [23]. 

Another feature of severe COVID-19 patients was lymphopenia and immunosuppression, which was responsible for hyperinflammation in the late stage of disease [29,30]. As the innate immune response alone may be insufficient for viral clearance, recruitment of lymphocytes appeared a more effective defense in virus infection. Alveolar capillary endothelial cells not only functioned as gas exchangers, but were also capable of recruiting immune cells through adhesion molecules and activating CD4^+^ T cells [31]. By binding to T cell integrin, ICAM1 increased T cell receptor (TCR) signaling to mediate the activation, adhesion, and migration of T cells [32,33]. Patients with COVID-19 showed pulmonary vascular injury associated with intracellular presence of SARS-CoV-2 and endothelial cell destruction [34]. In a SARS-CoV-2-infected vascularized lung-on-chip model, despite unproductive viral replication, lower CD31 expression and decreased barrier integrity were observed [35]. This evidence indicated endothelial injury might lead to impaired immune cell recruitment and thus increased hyperinflammation in the lung. In our study, SARS-CoV-2-infected hamsters had decreased gene expression levels of adhesion molecules (*Icam1* and *Vcam1*), suggesting suppression of endothelial cells’ activation. NOS3 was mainly expressed in endothelial cells, and NOS3-derived NO was involved in maintaining vascular homeostasis, through vasodilation, inhibition of vascular inflammation, and preventing endothelial cells apoptosis [36]. Together with NOS2, endogenous NO also produced regulated T cell differentiation and activation [37]. Our data showed that butyrate reversed the expression of *Nos3* and *Nos2* induced by SARS-CoV-2. Thus, butyrate offered endothelial protection and promoted endothelial cells’ activation.

Further exploring the effect of butyrate on reducing tissue damage, we observed that butyrate had anti-oxidative and anti-apoptotic effects. Increasing evidence suggests that pathological responses in COVID-19 patients are probably caused by oxidative stress [38]. Excessive ROS and subsequent MDA, a lipid peroxidation product, were both oxidative markers [39,40]. Moreover, there is decreased expression of the antioxidant enzyme SOD3 in the lungs of elderly COVID-19 patients [41]. Not only that, the link between oxidative stress and apoptosis has been proven [42]. Apoptosis induced by SARS-CoV-2 was associated with disease severity, and inhibition of intrinsic apoptosis could markedly ameliorate lung damage in transgenic mice that expressed human angiotensin-converting enzyme 2 (hACE2) [43,44]. BCL2 is known to suppress apoptosis by regulating ROS levels in the cytoplasm and mitochondria [42]. Thus, our findings suggested that butyrate might inhibit SARS-CoV-2-induced apoptosis by improving antioxidant capacity in the lung.

The expression of ACE2 in enterocytes makes the small intestine and colon more susceptible to SARS-CoV-2 infection, as it serves as the receptor for the virus [45,46]. By binding to ACE2, SARS-CoV-2 could cause direct damage to the intestinal epithelial barrier and activate the pro-inflammatory immune response to promote intestinal inflammation [47]. On the other hand, SARS-CoV-2-induced downregulated expression of ACE2 may lead to colon injury [48,49]. ACE2 can also function as a microecological modulator, regulating intestinal homeostasis, gut microbial ecology and innate immunity [50]. Wild-type mice receiving altered ileocecal microbiota transplantation from *Ace2* mutant mice were more likely to develop colitis, with increased infiltration of inflammatory intestinal bleeding and crypt damage. Dietary tryptophan can revert such microbiota change and rescue severe colitis in *Ace2*-deficient mice, suggesting a crosstalk between nutrition with this amino acid and innate immunity via ACE2 [50]. Notably, it has been found that butyrate could not only reduce expression of ACE2 and other genes essential for SARS-CoV-2 infection in gut epithelial organoids from rats, but also upregulate those involved in the antiviral pathways [23]. This again suggests a link between dietary nutrition and the anti-COVID-19 activity associated with ACE2.

In summary, we demonstrated that 500 mmol/L of butyrate supplemented in drinking water protected against SARS-CoV-2-induced tissue damage in golden hamsters. Among respiratory diseases, butyrate has previously been associated with regulation in chronic pulmonary disorders, but no significant effects were observed in treating SARS-CoV-2-infected hamsters with a combination of SCFAs (i.e., 200 mmol/L sodium acetate, 50 mmol/L propionate and 20 mmol/L butyrate, a concentration much lower than what we used in this study) [51,52]. Our study highlights the beneficial effects of butyrate on boosting antiviral immune response and reducing oxidative stress to promote cell survival in the disease.

## 4. Materials and Methods

### 4.1. Virus

The SARS-CoV-2 D614G variant AP62 (hCoV-19/China/AP62/2020, GISAID accession No. EPI_ISL_2779638) was used in this study. Virus stocks were prepared by three passages in Vero (ATCC CCL-81) cells in Dulbecco’s modified Eagle Medium (DMEM) (Gibco, Thermo Fisher Scientific, Waltham, MA, USA) with 1% penicillin–streptomycin (Gibco, Thermo Fisher Scientific, Waltham, MA, USA). Virus titers were measured using a plaque assay.

### 4.2. Experimental Animal and Study Design

Briefly, 8–10-week-old male golden hamsters were derived from Charles River Laboratories (Beijing Vital River Laboratory Animal Technology Co., Ltd., Beijing, China) and raised at specific pathogen-free animal feeding facilities. For the butyrate-treated group, sodium butyrate (Sigma-Aldrich, St. Louis, MO, USA) was supplemented in the drinking water at a final concentration of 500 mmol/L 12 days prior to virus inoculation and until the end of the experiment (3 or 5 dpi) (Figure 1). This dose was previously reported to improve the immune response of mice to influenza infection [53]. Control hamsters were supplied with water without butyrate during the experiment. Hamsters were anaesthetized with isoflurane and intranasally inoculated with a dose of 1 × 10^4^ plaque-forming units (PFU) of SARS-CoV-2 diluted in 200 μL phosphate-buffered saline (PBS; Gene Tech Shanghai Company Limited, Shanghai, China). Mock animals were inoculated with 200 μL PBS. The body weight of each hamster was measured daily during the course of the experiment. At 3 and 5 dpi, three and eight hamsters were euthanized, respectively. Blood samples were collected to prepare plasma. After gross observation and pathological examination, the trachea, lungs and colon were collected to determine the viral load or levels of host gene expression. Lung and colon tissues were also fixed in 10% formalin for histologic analysis. All experiments with the infectious virus were performed in biosafety level 3 (BSL-3) and animal biosafety level 3 (ABSL-3) containment facilities. The animal experiment was approved by the Medical Animal Care and Welfare Committee of Shantou University Medical College (Ref No. SUMC2023-058). 

### 4.3. Determination of Viral Load

Fresh trachea, lung and colon tissues (about 100 mg each sample) were collected and homogenized in PBS (100 mg/mL), respectively, and RNA was extracted using an RNA Extraction Kit (Wantai Beijing, Beijing, China). Quantitative real-time PCR (RT-qPCR) was performed to detect the ORF1ab and N genes of SARS-CoV-2 using a SARS-CoV-2 RT-qPCR Kit (Wantai Beijing, Beijing, China) on a SLAN-96S real-time PCR system (Hongshi Shanghai, Shanghai, China). Measurement of viral titer was carried out by standard TCID_50_ method in Vero cells. 

### 4.4. Determination of Host Gene Expression Level

Tissues kept in Invitrogen^TM^ RNAlater^TM^ Stabilization Solution (Thermo Fisher Scientific, Waltham, MA, USA) were homogenized in buffer RLT Plus (Qiagen, Hilden, Germany). Total RNA from lung and colon samples (around 30 mg per sample) was extracted using RNeasy Plus Mini Kit (Qiagen, Hilden, Germany) according to the manufacturer’s instructions. For determination of target gene expression level, cDNA was synthetized from the total RNA (no more than 5 μg, as instructed by the manufacturer) using a PrimeScript II 1st Strand cDNA Synthesis Kit (Takara, Dalian, China) and amplified using ChamQ Universal SYBR qPCR Master Mix (Vazyme Biotech, Nanjing, China) on a SLAN-96S real-time PCR system (Hongshi Shanghai, Shanghai, China). Primer sequences used for amplification are listed in Appendix A. For each sample, the host gene expression level was normalized to the housekeeping gene gamma-actin gene (*Actg*), and calculated using 2^−ΔΔCt^ method. 

### 4.5. Histologic Analysis

After being fixed in 10% formalin, lung and colon tissues were then embedded into paraffin and sectioned into 3–5 μm slices. The fixed lung sections were stained with hematoxylin and eosin (H&E) for histopathological analysis. Lung injury was evaluated according to pathological changes as follows: (1) a widened alveolar septum and consolidation; (2) pulmonary hemorrhage and edema; and (3) inflammatory cell infiltration. Each pathological change was scored on a scale of 0 to 4: 0 = no damage; 1 = mild injury; 2 = moderate injury; 3 = severe injury; and 4 = very severe injury [54]. For one lung lobe, the pathological score was the sum of these pathological changes. For each hamster, the comprehensive pathological score was averaged over the pathological score of three or four lung lobes. To elucidate the distribution of viral antigen in lung tissues, IHC was used to detect the NP of SARS-CoV-2. Briefly, a murine anti-SARS-CoV-2 NP specific monoclonal antibody (15A7-1, provided by Xiamen University) was applied as the primary antibody [54], and its binding to the goat anti-mouse IgG–biotin conjugate secondary antibody (BOSTER Biological Technology, Wuhan, China) was further labeled with horseradish peroxidase (HRP; MXB Biotechnologies, Fuzhou, China). The specific viral antigen was then visualized using 3,3′-diaminobenzidine (DAB; MXB Biotechnologies, Fuzhou, China) as the substrate. The colon sections were stained with Alcian Blue (AB) to analyze goblet cells and crypt length [55]. For each hamster, the number of goblet cells and the crypt length were averaged over at least five well-defined crypts [55]. Images were taken using an Axio Imager A2 microscope (Carl ZEISS, Jena, Germany). 

### 4.6. Apoptosis Assay

Formalin-fixed lung sections (3–5 μm slices) were stained with Hoechst 33,258 (Beyotime, Shanghai, China), and the images were captured using a fluorescence microscope (Axio Imager A2; Carl ZEISS, Jena, Germany). Apoptotic cells are characterized by condensed chromatin, so the cells that showed higher fluorescence intensity in nuclei were considered Hoechst-positive cells. The percentage of Hoechst-positive cells was measured in ImageJ version 1.53t (National Institute of Health, Bethesda, MD, USA).

### 4.7. Malondialdehyde (MDA) and Superoxide Dismutase (SOD) Assays

The prepared plasma (100 μL) was used for MDA measurement with a Lipid Peroxidation MDA Assay Kit (Beyotime, Shanghai, China). Freshly collected lung samples (about 60 mg per sample) were homogenized in 600 μL lysis buffer (Beyotime, Shanghai, China) for SOD detection using a Superoxide Dismutase (SOD) Assay Kit (Nanjing Jiancheng, Nanjing, China). One unit of SOD is defined as the amount of enzyme that causes 50% inhibition of the reduction reaction between water-soluble tetrazolium salt-1 (WST-1) and superoxide anion. 

### 4.8. Statistics

All results were presented as mean ± standard deviation (SD). A Student’s *t* test (two tailed) and a two-way analysis of variance (ANOVA) was used for the comparison of treatment groups. A Mann–Whitney test was used to calculate the *p* value in the case of non-normal distribution of data. *p* < 0.05 was statistically significant. **p* < 0.05, ***p* < 0.01, *** *p* < 0.001, ns = not significant. Graph generation and statistical analysis were performed in GraphPad Prism 8.0.1 (GraphPad Software, San Diego, CA, USA).

## Figures and Tables

**Figure 1 ijms-24-14191-f001:**
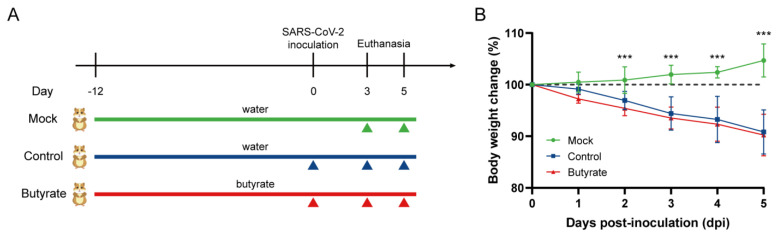
Body weight change and viral load in golden hamsters intranasally challenged with severe acute respiratory syndrome coronavirus 2 (SARS-CoV-2). (**A**) Study design. Hamsters were supplied with or without butyrate in their drinking water from day 12 prior to virus inoculation until the endpoint of the experiment. Mock animals were hamsters that received pure drinking water and no virus inoculation with SARS-CoV-2. Control indicated hamsters that received drinking water and intranasal inoculation of 1 × 10^4^ plaque-forming units (PFU) of SARS-CoV-2. Butyrate indicated hamsters receiving 500 mmol/L of sodium butyrate supplemented in their daily drinking water and SARS-CoV-2 inoculation. At days 3 (n = 3) and 5 (n = 8) post-inoculation (dpi), hamsters were euthanized, and samples were collected for further analysis. (**B**) Body weight change after virus inoculation. (**C**) Viral RNA and (**D**) viral titer detected in the trachea, lungs and colon of hamsters at 3 and 5 dpi. Data are represented as mean ± standard deviation (SD). Statistical significance was analyzed with a two-way analysis of variance (ANOVA). * *p* < 0.05, ** *p* < 0.01, *** *p* < 0.001, ns: not significant. The body weight change, viral RNA, and viral titer showed no significant difference between butyrate-treated hamsters and the control.

**Figure 2 ijms-24-14191-f002:**
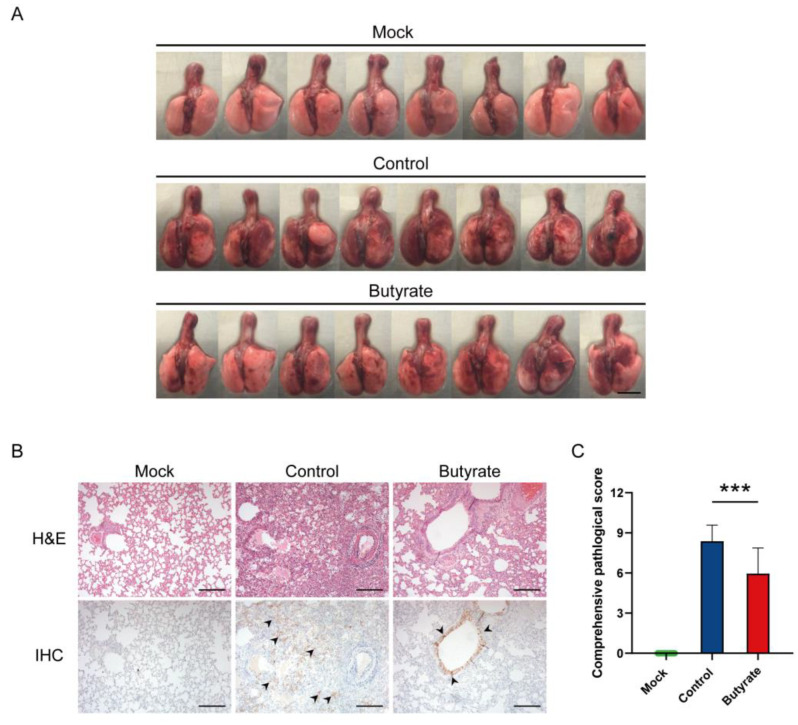
Pathological changes in the lung of golden hamsters intranasally inoculated with SARS-CoV-2. (**A**) Gross lung images of hamsters at 5 dpi. Scale bars, 1 cm. (**B**) Histopathological examination of the lungs at 5 dpi. Detection of SARS-CoV-2 nucleocapsid protein (NP)-positive cells visualized with brown coloration and indicated by black arrows. Scale bars, 200 μm. (**C**) Comprehensive pathological scores of the lungs at 5 dpi. Data are represented as mean ± SD. Statistical significance were analyzed with Student’s *t* test. *** *p* < 0.001.

**Figure 3 ijms-24-14191-f003:**
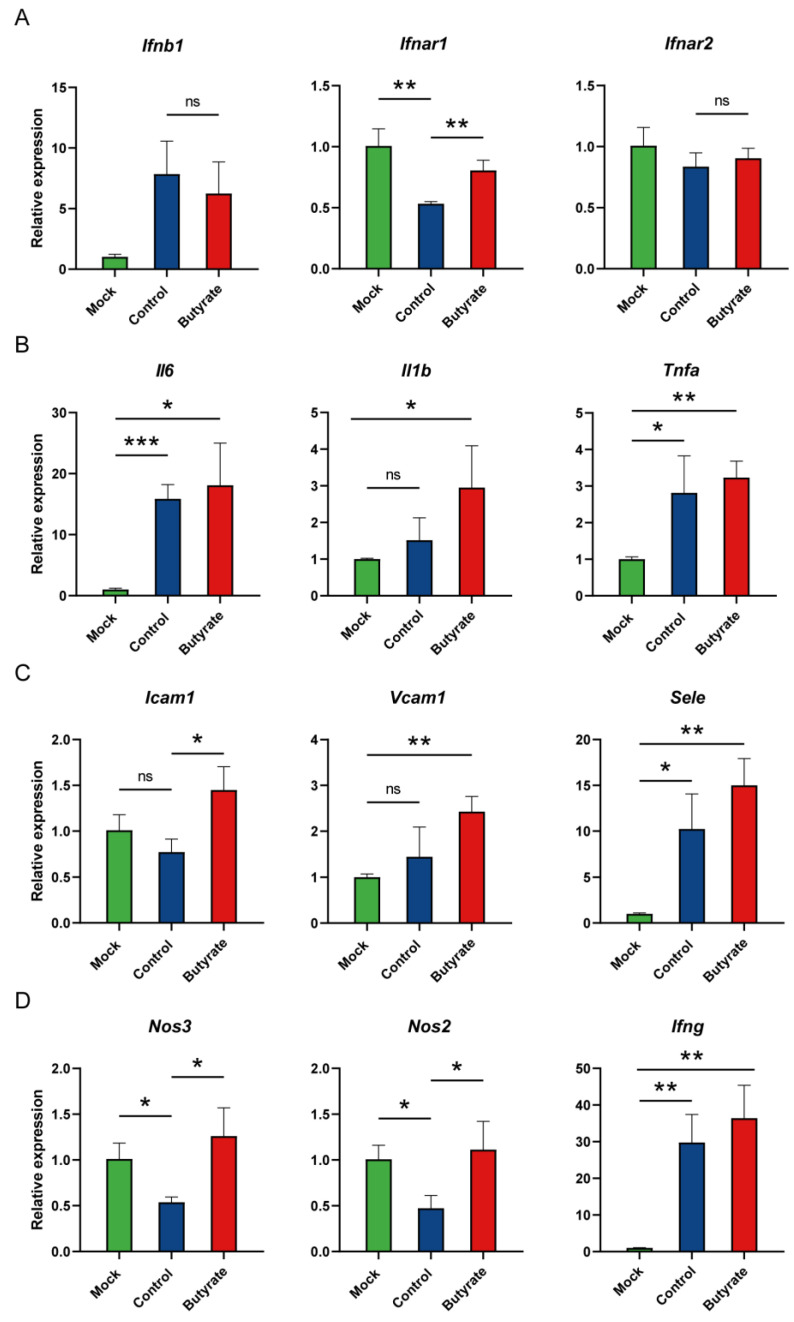
Expression levels of representative genes in golden hamsters intranasally challenged with SARS-CoV-2. Relative mRNA expression for representative genes in (**A**) type I interferon (IFN) signaling, (**B**) the proinflammatory effect, (**C**) endothelial cells’ activation and (**D**) nitric oxide production in the lungs at 5 dpi. The mRNA level was normalized to the housekeeping gene gamma-actin (*Actg*), and calculated using a 2^−ΔΔCt^ method. Data are represented as mean ± SD. Statistical significance was analyzed with Student’s *t* test. * *p* < 0.05, ** *p* < 0.01, *** *p* < 0.001, ns: not significant.

**Figure 4 ijms-24-14191-f004:**
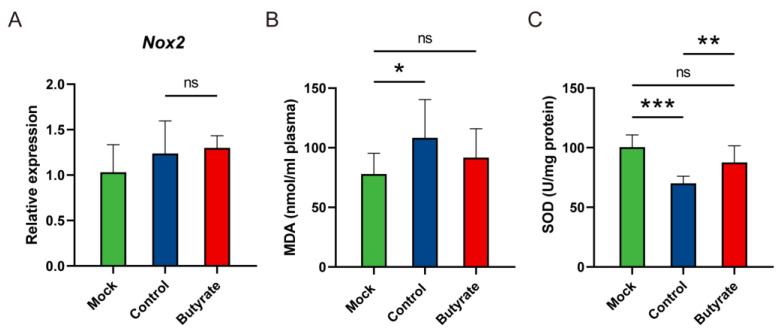
Oxidative status in golden hamsters intranasally challenged with SARS-CoV-2. (**A**) Relative mRNA expression for NADPH oxidase 2 (*Nox2*) in the lungs at 5 dpi. Data were normalized to the housekeeping gene gamma-actin (*Actg*) and calculated using a 2^−ΔΔCt^ method. (**B**) Malondialdehyde (MDA) levels in the plasma at 5 dpi. (**C**) Superoxide dismutase (SOD) activity in the lungs at 5 dpi. Data are represented as mean ± SD. Statistical significance was analyzed with Student’s *t* test. * *p* < 0.05, ** *p* < 0.01, *** *p* < 0.001, ns: not significant.

**Figure 5 ijms-24-14191-f005:**
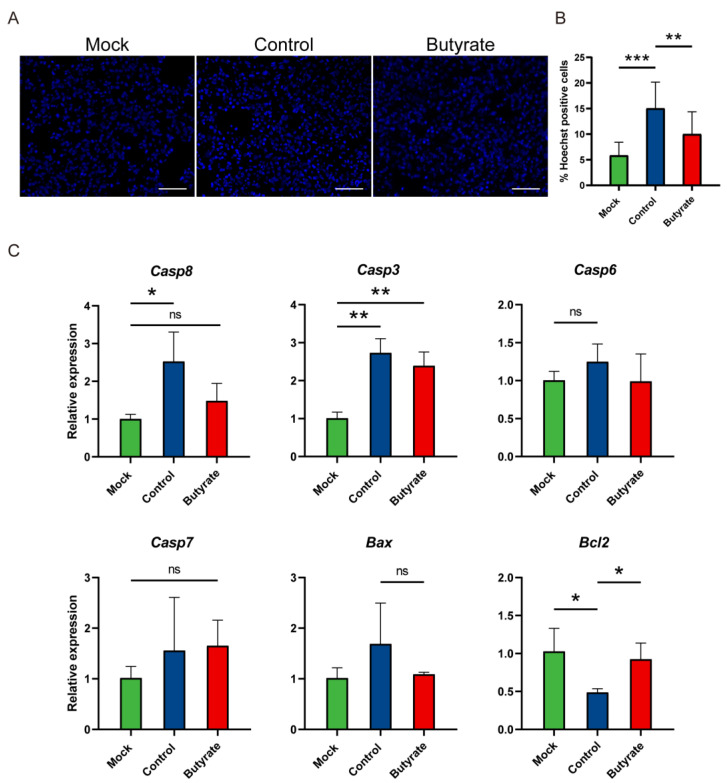
Apoptosis in golden hamsters intranasally challenged with SARS-CoV-2. (**A**) Hoechst staining of the lungs at 5 dpi. Scale bars, 50 μm. (**B**) Quantification of Hoechst-positive cells (cells with higher fluorescence intensity in nuclei) in the lungs. (**C**) Relative mRNA expression for representative genes in apoptosis pathways in the lungs at 5 dpi. The mRNA level was normalized to the housekeeping gene gamma-actin (*Actg*) and calculated using the 2^−ΔΔCt^ method. Data are represented as mean ± SD. Statistical significance was analyzed with Student’s *t* test. * *p* < 0.05, ** *p* < 0.01, *** *p* < 0.001, ns: not significant.

**Figure 6 ijms-24-14191-f006:**
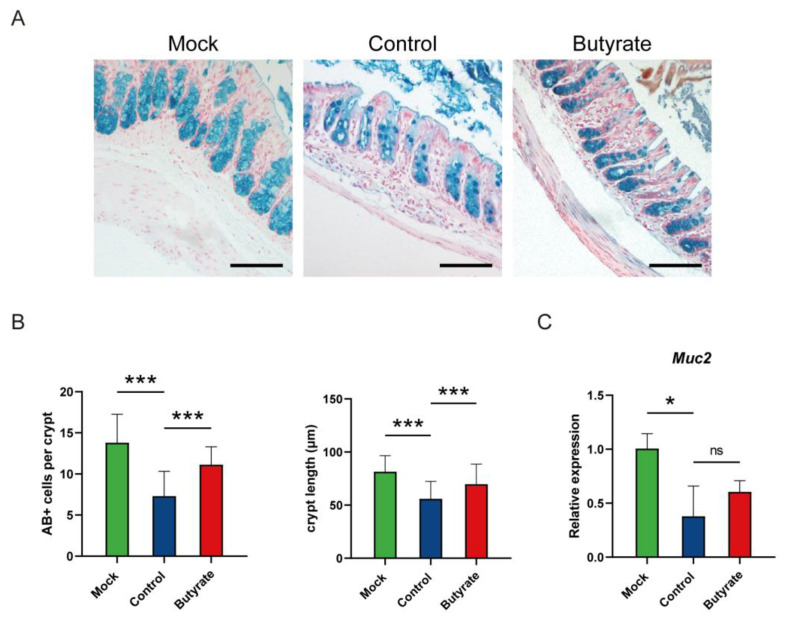
Goblet cells and *Muc2* expression in the colon of golden hamsters intranasally challenged with SARS-CoV-2. (**A**) AB-stained sections of colon. Scale bars, 100 μm. (**B**) Quantification of AB+ cells and crypt length in the colon at 5 dpi. (**C**) *Muc2* expression in the colon at 5 dpi. Data are represented as mean ± SD. Statistical significance was analyzed with Student’s *t* test. * *p* < 0.05, *** *p* < 0.001, ns: not significant.

## Data Availability

All data are available in the manuscript and Appendix A.

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
