# Peer review of "Butyrate Protects against SARS-CoV-2-Induced Tissue Damage in Golden Hamsters"

_ijms, 2023, doi:10.3390/ijms241814191_

Round 1
Reviewer 1 Report
Yu and coworkers revealed the effect of butyrate on lung and colon tissue in golden hamsters. Mainly, they demonstrated that butyrate alleviated lung injury in severe acute SARS-CoV-2-infected golden hamsters with supplementation of butyrate before and during the infection. Before publication, some additional discussions and quality of figures without additional experiment should be included.
1) In Figure 1B, representative image of IHC by SARS-CoV-2 NP staining in butyrate-treated group is stronger than that of control. Please explain the result, precisely. Additionally, the authors should arrange the region of the signal to move the center. Figure 1C data is statistical significance at P<0.001?
2) For, butyrate-treated group, sodium butyrate was supplemented in the drinking water at a final concentration of 500 mmol/L for 12 days. Simply, butyrate reaches to lung and stays for a long time?
3) In Figure 5C, the expression pattern of Casp6 and 7 had no effect. Please explain the negative data in the result of 2.5. (Apoptosis in hamsters).
4) It has been shown that ACE2 links amino acid malnutrition to microbial ecology and intestinal inflammation. SARS-CoV-2 binds to ACE2. The data concerning colon in hamster are just "2.6. goblet cells and Muc2 expression in hamsters". Please discuss colon damage occurred by SARS-CoV-2 infection, using the reference (Hashimoto et al. Nature, 2012, 487, 477-481) and others for obtaining fruitful discussion.
5) No significant effects are observed in treating SARS-CoV-2 infected hamsters with a combination of SCFAs, but significant effects of just butyrate in the manuscript. Please explain the reason why just butyrate has effects in lung and colon in hamsters in detail.
6) In Materials and Methods, "4.6. Apoptosis assay", the only Hoechst positive cell measurement is sufficient for apoptosis assay?
Author Response
Response to the reviewer’s comments:
Yu and coworkers revealed the effect of butyrate on lung and colon tissue in golden hamsters. Mainly, they demonstrated that butyrate alleviated lung injury in severe acute SARS-CoV-2-infected golden hamsters with supplementation of butyrate before and during the infection. Before publication, some additional discussions and quality of figures without additional experiment should be included.
1) In Figure 1B, representative image of IHC by SARS-CoV-2 NP staining in butyrate-treated group is stronger than that of control. Please explain the result, precisely. Additionally, the authors should arrange the region of the signal to move the center. Figure 1C data is statistical significance at P<0.001?
Response: Representative images of IHC were shown in Figure 2B. NP staining is stronger in intensity in some specific regions of butyrate-treated group, but less in amount across the whole lung section (i.e. expressed in higher levels but only limited in the epithelial cells of bronchioles or the peribronchiolar pulmonary cells), when compared with the control group.
We thank the reviewer for raising this point and have added a paragraph describing and explaining this observation:
“Immunohistochemistry (IHC) for SARS-CoV-2 N Protein (NP) detection indicated that specific viral antigens were widely expressed and distributed in the alveolar cells and pulmonary exudate of each control animal. In contrast, NP-positive cells were majorly observed in the epithelial cells of bronchioles in the butyrate-treated hamsters (Figure 2B). These distinct viral antigen distribution patterns (Figure 2B), rather than differences in the viral load (Figure 1C, D), suggested limited virus dissemination in the bronchioloalveolar tissues following butyrate treatment, which in turn protected against SARS-CoV-2 by reducing tissue destruction (Figure 2A, B) and overall pathology (Figure 1B).” (Lines 103-111, pages 3-4)
Additional information has also been provided to describe the IHC method:
Figure 2: “(B) Histopathological examination of the lungs at 5 dpi. Detection of SARS-CoV-2 NP-positive cells are visualized as brown coloration and indicated by black arrows. Scale bars, 200 μm.” (Lines 115-117, page 4)
“Briefly, a murine anti-SARS-CoV-2 NP specific monoclonal antibody (15A7-1, provided by Xiamen University) was applied as primary antibody [54], and its binding to the goat anti-mouse IgG–biotin conjugate secondary antibody (BOSTER Biological Technology) was further labeled with horseradish peroxidase (HRP; MXB Biotechnologies). Specific viral antigen was then visualized by using 3,3′-diaminobenzidine (DAB; MXB Bio-technologies) as the substrate.” (Lines 335-340, page 12)
Figures were adjusted to show the signals more clearly in the center (Figure 2B, page 4).
We confirmed that data of Figure 2C (comprehensive pathological scores of the lungs) was statistically significant at P<0.001, while Figure 1C (viral RNA load) was not (P>0.05).
2) For butyrate-treated group, sodium butyrate was supplemented in the drinking water at a final concentration of 500 mmol/L for 12 days. Simply, butyrate reaches to lung and stays for a long time?
Response: When sodium butyrate is supplemented in the drinking water, it can be absorbed into the blood circulation through the gastrointestinal tract, as reported previously in humans and in laboratory animals (DOI: 10.1038/s41598-018-37246-7, 10.1038/s41467-022-35767-4, 10.1021/acs.jafc.7b04666, 10.1016/j.metabol.2017.02.003), but data was not available on butyrate distribution in lungs following oral administration. A few recent reports (DOI: 10.1152/ajplung.00421.2020, 10.1164/rccm.201909-1840LE) found short-chain fatty acids in human lung tissue and bronchoalveolar lavage fluid (BALF) of healthy people, of which butyric acid only accounted for <5% and its concentration was very low (about 1-5 μM).
Although there might be some minimal levels of butyrate in the lungs after absorption into the blood stream, lungs are not previously considered a target site of accumulation. Moreover, as butyrate is metabolized relatively quickly in the body, its presence in the lungs may not be long-lasting. For the protection against respiratory infections and the changes of disease outcome, gut-lung communication and regulation in the immune response may be more plausible explanations. However, as research on butyrate supplementation and its effects on the respiratory system is still ongoing, and metabolism and clearance of butyrate can vary among individuals and depend on various factors such as metabolism rate, overall health, and specific metabolic pathways, further investigation is warranted to fully understand the dynamic distribution, metabolism, and the exact functional mechanisms of butyrate, including its presence and duration in the lungs.
3) In Figure 5C, the expression pattern of Casp6 and 7 had no effect. Please explain the negative data in the result of 2.5. (Apoptosis in hamsters).
Response: Caspases regulate cell death, immune responses, and homeostasis, and their sequential activation plays a central role in the execution-phase of cell apoptosis. Caspases -3, -6 and -7 are categorized as executioner caspases but caspase 6 can also profoundly mediate innate immunity, inflammasome activation, and host defense against some virus infections (doi: 10.1016/j.cell.2020.03.040.). The precursor of Caspases 6 and 7 would be activated upon stimuli and by other caspases such as caspases 3, 8, 9 and 10. It is noted that caspases 8 and 9 are upstream caspases that can process and activate caspase 3, while caspase 3 cleaves and activates caspases 6 and 7. Therefore, caspases 6 and 7 are relatively downstream in the regulation of apoptosis. In this study, we only measure the RNA expression level of some caspases at 5 dpi and found butyrate-treated hamsters had less apoptotic cells (Figures 5A, 5B). This may not be the optimal time point for the RNA expression of all the caspases, whose sequential activation and signaling pathway should be further explored.
4) It has been shown that ACE2 links amino acid malnutrition to microbial ecology and intestinal inflammation. SARS-CoV-2 binds to ACE2. The data concerning colon in hamster are just "2.6. goblet cells and Muc2 expression in hamsters". Please discuss colon damage occurred by SARS-CoV-2 infection, using the reference (Hashimoto et al. Nature, 2012, 487, 477-481) and others for obtaining fruitful discussion.
Response: We thank the reviewer for the insightful comments and suggestions. A paragraph has been added to the Discussion (Lines 255-270, page 10):
“The expression of ACE2 in enterocytes makes the small intestine and colon more susceptible to SARS-CoV-2 infection, as it serves as the receptor for the virus [45, 46]. By binding to ACE2, SARS-CoV-2 could cause direct damage to intestinal epithelial barrier and activate pro-inflammatory immune response to promote intestinal inflammation [47]. On the other hand, SARS-CoV-2-induced downregulated expression of ACE2 may lead to colon injury [48, 49]. ACE2 can also function as a microecological modulator, regulating intestinal homeostasis, gut microbial ecology and innate immunity. Wild-type mice receiving altered ileocaecal microbiota transplantation from Ace2 mutant mice were more likely to develop colitis, with increased infiltration of inflammatory, intestinal bleeding and crypt damage. Dietary tryptophan can revert such microbiota change and rescue severe colitis in the Ace2 deficient mice, suggesting a crosstalk between this amino acid nutrition and the innate immunity via ACE2 [50]. Notably, it has been found that butyrate could reduce expression of ACE2 and other genes essential for SARS-CoV-2 infection in gut epithelial organoids from rats, but also upregulating those involved in the antiviral pathways [23]. This again suggests a link between the dietary nutrition and the anti-COVID-19 activity associated with ACE2.”
5) No significant effects are observed in treating SARS-CoV-2 infected hamsters with a combination of SCFAs, but significant effects of just butyrate in the manuscript. Please explain the reason why just butyrate has effects in lung and colon in hamsters in detail.
Response: The treatment effects of SCFAs significantly rely on the selection of animal models and the therapeutic regimen. Changes in the viral dose or strain, inoculation routes, gender, age or health conditions of hamsters, butyrate or SCFA dose or treatment regimens, may all affect the disease outcome.
Here we used a D614G variant of SARS-CoV-2 which can cause severe acute pneumonia in 8-10-week-old male golden hamsters within a week when intranasally inoculated at a sublethal dose of 1×104 PFU. This model has been well-established in our lab (DOI: 10.1002/advs.202207249 and Reference [54]) and is sufficient to observe the differential pathological changes after dietary supplement treatment. Sodium butyrate was supplemented in the drinking water at a final concentration of 500 mmol/L 12 days prior to virus inoculation and until the end of the experiment, a dose previously reported to improve the immune response of mice to influenza infection (Reference [53]).
Sencio et al (Reference [52]) used a similar D614G variant to show that a combined SCFA supplementation in drinking water had no effect on clinical and inflammatory parameters in hamsters nasally inoculated with 2 × 104 TCID50 of SARS-CoV-2, a dose comparable to what we used in this study. However, their animals were only treated five days before and during infection, with a combination of sodium acetate (200 mM), propionate (50 mM) and butyrate (20 mM). This pretreatment period was much shorter than ours, while the SCFA concentration was also lower, which could lead to the discrepancy in the overall effects on disease outcomes.
To highlight the difference in the oral administration dose of butyrate, we modified the final paragraph in the Discussion:
“In summary, we demonstrated that 500 mmol/L of butyrate protected against SARS-CoV-2-induced tissue damage in golden hamsters. Among respiratory diseases, butyrate has previously been associated with regulation in chronic pulmonary disorders but no significant effects were observed in treating SARS-CoV-2-infected hamsters with a combination of SCFAs (i.e. 200 mmol/L sodium acetate, 50 mmol/L propionate and 20 mmol/L butyrate, a concentration much lower than what we used in this study) [51, 52]. Our study highlighted the beneficial effects of butyrate on boosting antiviral immune response and reducing oxidative stress to promote cell survival in the disease.” (Lines 271-278, page 10)
6) In Materials and Methods, "4.6. Apoptosis assay", the only Hoechst positive cell measurement is sufficient for apoptosis assay?
Response: In this study, detection of Hoechst positive cells and changes in some apoptosis-related genes such as caspases and Bcl2 suggested evidence of apoptosis. We fully agree that further investigation is needed to confirm the results and to reveal the underlying mechanisms for butyrate-related inhibition in cell death.
Reviewer 2 Report
The paper of Y. Guan and H. ZHu is dedicate to a very interesting aspect of viral infections, namely to immunorestponse to SARS-Cov2 and the role of butyrate in that response. Paper is well scheduled, interestingly written and sound and shopuld be published as it is.
In my copy of the paper Figure 5A is non-informative since it is to dark and the chnges are non-visible. However, this could be corrected upon proofreading procedure.
Author Response
We thank the reviewer for the comments and high-resolution figures will be separately uploaded to the website upon submission of the revised manuscript.
Reviewer 3 Report
The present manuscript describes the properties of butyrate against SARS-CoV-2-infected golden hamsters in upregulating antiviral immune responses and promoting cell survival. The study is well-conducted and the article properly written. However, some references are missing and introduction section should be implemented. Many studies describing anti-SARS-CoV-2 effect of butyrate and, in general, its antiviral activity should be cited and added in the introduction. Below are some examples:
doi: 10.1128/JVI.00326-20, doi: 10.3390/molecules27030862,
DOI: 10.1038/s41598-022-11122-x, DOI: 10.1161/HYPERTENSIONAHA.120.16647
Author Response
We thank the reviewer for the reminder and suggestions. References on the antiviral effects of butyrate against SARS-CoV-2 have been included in the Introduction (Lines 57-58, page 2) and Discussion (Lines 218-220, page 9; Lines 266-270, page 10).
Reviewer 4 Report
This study is well-planned, and the manuscript is clearly written and easy to follow. However, there are still some aspects that need to be clarified and improved. First, there is no justification and scientific support for the applied dosage of butyrate – please comment on this in the manuscript. Secondly, there is a lack of butyrate concentration in the plasma or tissue. Was it measured in this study or do Authors have knowledge from a pilot study on this?
Minor remarks
Line 278 Please provide the molar concentration of PBS components, if different than usually used.
Line 288 please provide the amount of total RNA taken for reverse transcription.
Line 317 please give the amount of lung tissue taken for the analysis.
Section 2.3 please correct the abbreviations of gene names by avoiding Greek symbols e.g. Tnfa
